# Taking Action towards an Inclusive Career Counselling for Asylum Seekers and Refugees—A Literature Review Based on the PRISMA Model

**DOI:** 10.3390/bs13120962

**Published:** 2023-11-22

**Authors:** Gresa Beqiraj, Lea Ferrari

**Affiliations:** Department of Philosophy, Sociology, Education and Applied Psychology, University of Padua, 35131 Padua, Italy; lea.ferrari@unipd.it

**Keywords:** career counselling, career development, refugees, asylum seekers, forced migrants, under-represented groups, actions, career adaptability, systems of influences

## Abstract

Over the past decade, scholarly attention has increasingly focused on what is known as the ‘refugee gap’, which refers to the great difficulty asylum seekers and refugees face in entering the labour market in the host country. This poses a grave threat of social and occupational marginalisation for this group and highlights the role of systemic factors in facilitating resilience outcomes. By adopting a systemic perspective, this research aims to provide a critical reflection on the key features that should be considered when designing and implementing effective career counselling interventions for asylum seekers and refugees. To this end, a systematic review of the international applied research published by October 2022 has been carried out, using the PRISMA model. The 20 selected publications are grouped according to three main career research strands that reflect the core aspects to be addressed within career interventions in order to assist asylum seekers and refugees in their life and career transition in the host country, namely (i) psychosocial resources, (ii) identity transformation processes, and (iii) lived experiences and meaning-making processes. The main findings of the studies are discussed by looking into common themes that emerge from the literature, namely challenges related to migration contexts, self-regulating personal resources and skills, and sense of self and identity in career transitions, as well as considerations on the design aspects of career counselling and research targeting As&R. Finally, some guidelines and directions for action are provided for the purpose of developing inclusive career counselling interventions for As&R.

## 1. Introduction

The ‘refugee crisis’ has been central to political and social debates in many countries over the past decade. According to UNHCR statistics on forced displacements, the number of refugees worldwide increased from approximately 27 million in 2021 to more than 35 million at the end of 2022, marking the largest increase ever recorded [1].

Evidence from the literature highlights the growing phenomenon of the so-called ‘refugee gap’, which is characterised by lower employment rates among asylum seekers and refugees (As&R) compared to other migrants and the native population, even long after arrival in the host country [2,3,4]. Regardless of job titles and ethnic, linguistic, or cultural factors, simply being identified as a ‘refugee’ results in discriminatory attitudes during recruitment processes by employers and in reduced job opportunities [4]. Such great difficulty in entering the labour market of the host country is associated with a high risk of social and occupational marginalisation resulting from a combination of factors. These include pre- and post-migration contexts, cultural diversity, and policy orientations at the international, European, or national level of the host countries [2,5,6]. Considering the complex interplay between these factors, a deeper insight into the existing psychological literature on career counselling and career development interventions focusing on As&R is crucial to identifying the assets that can promote resilience outcomes.

Thus, this paper outlines and discusses the key features considered in the empirical career psychology literature targeting As&R that should be taken into account when designing and implementing effective career counselling interventions aimed at mitigating the refugee gap phenomenon. To this end, we carry out a systematic review of international empirical studies and apply theoretical research published up until October 2022, adhering to the PRISMA model.

As&R, the target population of this research study, includes the category of ‘refugees’ and ‘asylum seekers’, referring to the right to asylum as acknowledged in the Universal Declaration of Human Rights and in the various forms of protection recognized by international law (i.e., the Geneva Convention of 28 July 1951 [7]), European law (i.e., Lisbon Treaty, 2007 [8]), and the domestic laws of the hosting countries. Rapid transformations in today’s world of work and the cultural and socio-economic heterogeneity of the target population have highlighted the inability of conventional theoretical models and applicative tools to accurately reflect the complexity of the reality of social and work contexts [9]. Recent literature draws attention to the need to extend the scope of career interventions beyond the individual level and suggests that the setting of career counselling should consider the dynamic interaction of individual, cultural, socio-economic, and political factors [10,11,12,13] while also taking advocacy actions.

In the analysis of the extant literature and in the subsequent development of career counselling guidelines, this review embraces a systemic conceptual viewpoint on career development according to McMahon et al. [9,11]. Such a theoretical framework considers individuals as open systems situated in intricate, fluid, and interrelated broad contexts of influence [9,11]. The individual system, which includes psycho-emotional and personal characteristics, is influenced by direct social interaction contexts that constitute the ‘social system’. The dynamic processes within and between these two systems occur within the broader ‘environmental-societal system’ of socio-economic, geo-political, and cultural contexts [9,11].

Referring to these systemic levels as a framework and criterion for career intervention design and implementation, this review aims to provide valuable suggestions for future theoretical advancements in career psychology. It also seeks to contribute to the development of career interventions that are responsive to intertwined life contexts and inclusive of underrepresented social groups.

Three main steps were taken to organise the review. The first step involved planning, aimed at defining the review protocol and research questions. The review conducted as a second step consisted of selecting the relevant studies, developing the PRISMA diagram, and defining the research methodology. The third step involved synthesising and discussing the results. It includes a descriptive analysis of the three main strands of research focusing on (i) psychosocial resources, (ii) identity transformation processes, and (iii) lived experiences and meaning-making processes and corresponding summary tables of the main characteristics of the studies, followed by an analysis of common themes and design aspects, as well as the main limitations of this review. Finally, the paper presents a visual overview of the analysis and discussion to inform career counselling practice and to guide future research. A graphical overview of the main resources and barriers in As&R career transitions is presented, together with a table of recommendations.

## 2. Materials and Methods

In the period between November 2022 and February 2023, we conducted a systematic literature review using the PRISMA model—Preferred Reporting Items for Systematic Reviews and Meta-Analyses [14,15,16]. This method is designed to systematically identify and analyse existing scientific contributions on a given topic through transparent and codified selection processes. The guidelines for undertaking the review report require an output of the results by means of a flowchart pertaining to the four stages of the process: (i) identification, (ii) screening, (iii) eligibility, and (iv) inclusion of studies [16]. The review was guided by the following research questions:-What practical insights can be gathered from the literature on the career development of As&R in hosting countries?-Which intervention features are recommended to support the career development of this population?

To address the research questions, a bibliographic search was conducted in the area of career psychology and career counselling. Eight relevant keywords were selected and searched for across three primary psychological databases—APA PsycInfo, Psychology and Behavioral Sciences Collection, and APA PsycArticles—via EBSCOhost. To conduct a thorough exploration of this emerging area of intervention, keywords were also searched in the multidisciplinary database Web of Science.

The first four keywords refer to the target population, namely refugees, asylum seekers, forced migrants, and forced immigrants. The remaining four concern the research domain, namely career counselling, career development, vocational guidance, and vocational counselling. To ensure a high relevance of the findings, the data were cross-referenced using the Boolean operator ‘AND’ to include a combination of at least one of the four target population keywords and one of the research domain keywords. No restrictions were imposed on the selection of search fields, types of publication, or the earliest publication period. The review encompassed publications from January 1900 until October 2022.

Given the practical aim of this research, solely empirical and applicative research studies were taken into account. As regards eligibility criteria, all studies had to refer to adult As&R (16 years old and above) and concern career development, career counselling, and career guidance interventions. Due to the distinctive characteristics of the target population in terms of pre- and post-migration life experiences as well as of specific social rights in the country of asylum, literature related to the broad category of ‘migrant’ or ‘immigrant’ was not included, unless there was an explicit reference to the As&R category.

Out of 206 identified publications, 19 appeared in more than one database (duplicates). Among the remaining 187 screened publications, 54 were fully assessed. Of these, 19 were ultimately included for review. After reviewing the bibliographical references of the 19 selected publications, one additional publication was deemed relevant to the present research and was subsequently included. All 20 selected publications were accessible.

The following publications were excluded: 94 studies that did not reflect the topic of this review and the field of career psychology or concerned the general migrant population but not adult As&R; 17 publications that were not related to or had implications for career interventions; 14 publications that focused on education or access to education for young As&R; 29 publications that dealt with mental health and psychological wellbeing issues; and 14 publications that dealt strictly with theoretical aspects or were secondary sources.

The literature selection process is depicted in the following diagram, which reproduces the steps involved in the PRISMA model (Figure 1).

Considering the limited number of studies, the diversity of study designs, and the heterogeneity of the As&R population studied, a descriptive analysis of the results was considered appropriate and in line with the purpose of the paper and its practical insights. For each of the selected studies, we present the main references (author(s), title, and year), the participants, the objective of the research or of the career intervention, its design and methodological characteristics, and the main results. The analysis of these elements enabled us to classify the studies and illustrate the three primary strands of research in career psychology that emerge from the selected literature. Subsequently, we carried out a qualitative analysis to investigate essential thematic and design aspects using a narrative, theme-focused approach. Both thematic and design aspects were classified using a systemic framework. In the various stages of the review process, the first author performed the initial study categorisation and later conferred with the second author, who holds specific expertise in the field, to address any uncertainties. To enhance the validity of the analysis, the outcomes were compared and incorporated with insights from the theoretical literature on career psychology, as well as multidisciplinary contributions.

## 3. Results

All 20 selected studies were published after 2010—3 in the period from 2011 to 2014, and the remaining 17 between 2018 and 2022. These studies cover a total of 1181 participants and were carried out in 11 different countries, mainly in Europe (13 out of 20 studies). As regards the target population of the selected studies, seven (7) studies focused on persons with refugee status, one (1) on asylum seekers, and twelve (12) used mixed samples or did not provide information on the legacy status. Ten (10) studies included As&R from specific geographic backgrounds, i.e., seven (7) of African origin and three (3) of Syrian origin, while the remaining study populations were from various other countries. Concerning gender, three (3) studies focused exclusively on women. The following chart depicts asylum countries and sample dimensions of the studies (Figure 2).

Many of the analysed publications [17,18,19,20,21,22,23,24,25] describe the results of quantitative or qualitative research studies on both contextual and individual factors and resources, which are considered to be helpful in fostering the professional development of As&R. Other publications [26,27,28,29] explore professional identity processes. The remaining studies [30,31,32,33,34,35,36] present, implement, or evaluate the effectiveness of career interventions or the specific aspects addressed by them. More specifically, these publications include individual or group-based career interventions and counselling activities. 

The studies are grouped in the sections below according to three main research strands that reflect the core aspects to be developed within career interventions: (i) psychosocial resources, (ii) identity transformation processes, and (iii) lived experiences and meaning-making processes. A table for each group summarises the main data of the publications, i.e., the study references and participants, the objective of the research or career intervention, its design and methodological features, and the main results.

### 3.1. First Group: Psychosocial Resources

Nine (9) studies [17,18,19,20,21,22,23,24,25] were included in the first group: five (5) adopted a quantitative design, three (3) a qualitative design, and one (1) a mixed-method design (Table 1). Research involved As&R who resided in Italy [17,20,22], Germany [18,23], Canada [24], Australia [25], Switzerland [21], Greece, and the Netherlands [19]. This group of studies examines career transitions, focusing on psychosocial resources, personal characteristics, and adaptive skills among As&R. This body of research supports the relevance of developing these skills and resources in career interventions as a means of addressing contextual constraints and supporting professional integration processes for As&R. Two studies assess the effectiveness of career interventions for As&R [17,24], while others investigate the role of attitudes and psychological characteristics enabling an agentic perspective.

The majority of the studies in this group encompass qualitative, quantitative, and mixed-method research [17,18,19,22,23,24] to examine the construct of ‘career adaptability’ using the Career Construction Model of Adaptation [37,38]. One qualitative study [25] draws on the conceptualisation of the construct by McMahon et al. [39]. The four coping strategies of career adaptability (concern, control, curiosity, and confidence), as identified by the Career Construction Model of Adaptation [37], are examined in relation to a range of individual resources and contextual barriers. The theoretical description of the construct, according to McMahon et al. [39], includes a fifth dimension, i.e., cooperation, which is explored in one qualitative research [25] in relation to dynamic systemic influences.

Other studies have identified additional personal resources that might be beneficial to the career transition process of AS&R, namely hope-action competencies (hope, self-reflection, self-clarity, visioning, goal setting, implementing, adapting, and job search clarity) [24]; courage [20]; entrepreneurial inclination [18]; and proactivity, professional hope, resilience, and optimism [24].

### 3.2. Second Group: Identity Transformation Processes

Four (4) studies [26,27,28,29] that investigate the professional development of As&R were included in the second group (see Table 2). The studies mainly concentrate on the process through which As&R transform their career identity by interacting with contextual barriers in the host countries. The four publications consist of qualitative studies that conducted semi-structured interviews with As&R residing in Germany [26,29], the United Kingdom [28], and South Africa [27]. The studies employ purposive snowball sampling methods based on the participants’ refugee status and their professional and qualification profiles. For instance, studies focus on refugees with recognized status possessing professional qualifications in medicine and education [28], individuals of Syrian origin with higher qualifications [26], exclusively female [27], and mixed As&R [29] with no additional qualification characteristics.

### 3.3. Third Group: Lived Experiences and Meaning-Making Processes

Seven (7) studies [30,31,32,33,34,35,36] were included in the third group (see Table 3). These involve As&R residing in Australia [30,31], Denmark [32], the United States [33], the United Kingdom [34], Greece [35], and Italy [36]. The studies have a shared understanding of professional development and career interventions for As&R. The focus is on the whole lifespan and multiple existential domains of individuals, as well as their attribution and meaning-making processes. The research examines single case studies or small snowball samples, emphasizing the significance of the participants’ lived experiences and the meaning they ascribe to them. The methodological tools used include semi-structured interviews, models, and narrative elicitation techniques. Creative representations such as drawings and graphics are also included.

## 4. Discussion

The key findings of the reviewed studies are discussed below according to two different criteria of analysis: thematic and design-related aspects. By drawing attention to interlinked critical themes and design-related aspects within the reviewed studies, the paper provides a more comprehensive overview of the relevant aspects of career and life transitions of AS&R. It also offers valuable insights that can be employed when designing and implementing future career interventions for As&R. A systemic lens is adopted to shed light on the influences and interrelationships among factors at the individual, social, and ‘environmental-societal’ system levels that are relevant to career and life transitions of AS&R [11]. It should be noted that the discussion of common themes and design features outlined below is not intended to be generalised and that, in practice, the considerations made should be situated within country- and context-specific dimensions.

### 4.1. Thematic Aspects

The main topics addressed in the studies are grouped into three main areas, referring to the challenges posed by migration contexts, the role of self-regulating personal resources and skills, and the sense of self and identity in As&R career transitions.

#### 4.1.1. Challenges Related to Migration Contexts

Notwithstanding the differences in migration pathways and asylum country-specific characteristics in terms of institutional service networks, procedures, and access to social rights, the experiences reported by As&R in the reviewed studies revealed common barriers and resources, which are also confirmed by further international psychological and multidisciplinary literature.

Regarding the migratory experience, the reviewed research draws attention to the impact of current migration and asylum policies. The factors considered include the perilous nature of migratory routes; the oppression, violation of basic human rights, violence, and abuse experienced [20,31,40,41,42]; and the effect of borders as ‘natural selection’, whereby crossing them depends on the economic, physical, and psychological resilience of As&R [19,41]. Indeed, worldwide restrictive asylum policies have focused on controlling migration routes and implementing refoulement measures, as well as on shifting geopolitical borders through regulatory changes that generate illegality and result in refugees being pushed back to countries (thanks to agreements with these ‘safe third countries’) where human rights violations take place [38,39]. Such practices result not only in dehumanizing and abusive migratory experiences [20,41,42] but they also exacerbate global inequalities and contribute to the widening of the North-South divide [19,41].

In light of this, migration narratives in career counselling interventions need to be contextualized within the influences of ‘social-environmental’ systems to raise As&R awareness on how their lived experiences are shaped by political, historical, and cultural contexts [11,31]. Indeed, geopolitical factors and migration policies impact As&R experiences at both the individual and social levels by making violence and dehumanizing experiences an anticipated condition and accepted price to pay [40]. Over time, this leads to negative cognitive, psychosocial, and emotional effects such as isolation, lack of control, confidence, and identification with disempowered self-perspectives that affect their self-esteem, mental, and physical health [20,28,29,31,40].

Similarly, regarding the post-migration phase, the studies reviewed outlined specific influences at the ‘environmental-societal’ level. These include procedures for recognition, providing accommodation, and access to social rights, services, and integration programmes [21,26,41,42,43]. Although these aspects differ across asylum countries, common features impacting As&R individual and social systems were identified. Specifically, the transition period in reception centres while awaiting recognition of refugee status leads asylum seekers to common experiences of ‘learned helplessness’ [18,20,23,29,31], feelings of suspension owing to the indefinite duration of asylum procedures and the temporary residence status [2,41,44,45,46,47,48], inactivity and inability to influence their personal and professional lives, and reduced confidence [2,18,28,29,47]. Another constraint is the disregard or lack of acknowledgement of professional qualifications and expertise, which was found to be detrimental to As&R need for value, distinctiveness, and continuity [29]. Furthermore, the impacts of the above-mentioned post-migration ‘environmental-societal’ contexts, which Phillimore and Cheung [48] define as ‘violence of uncertainty’, prevent As&R from safeguarding their families’ livelihoods and protecting themselves, and it undermines their sense of belonging, connectedness, and social recognition [29]. These findings are consistent with existing clinical psychology literature investigating the effects of post-migration management policies and other stressors on the mental and physical health of As&R [46,47,48], with the effects being particularly severe for women and persisting beyond 21 months after being granted refugee status [48].

While the majority of the reviewed literature concentrates on contextual difficulties, the aid provided by social and institutional networks for refugee status recognition has proven to be of great benefit to As&R [21,29]. Moreover, career counselling, psychological and spiritual support, and social networks have proven to be particularly valuable in helping As&R to share their difficulties, give meaning to their existence, explore positive self-narratives and strategies, and connect with their environment in a functional way [29,32,34].

It is worth noting that when looking at the key pre- and post-migration ‘environmental-societal’ factors, their influence on the individual and social system level appears to be top-down. There is no evidence suggesting a bottom-up agency on these macro-level factors on the part of AS&R and their closest networks. Therefore, it is crucial for career practitioners to extend career interventions beyond individually focused counselling and consider the dynamic interplay of systemic influences in order to actively campaign against policies that exclude and violate the basic rights and needs of As&R [20]. To this end, practitioners need to develop interdisciplinary skills to better analyse the interrelated influences of As&R life contexts. Career counsellors should also adopt attitudes and develop tools that help to create a trusting and open relational environment and reduce the inequality, fear, and mistrust that As&R may experience as a result of difficult and abusive migration experiences [31,32,33,35,49]. Furthermore, career psychology ought to strengthen its political agency by disseminating psychosocial reports on As&R and the impact of macro-level factors like the common determinants of migration and migration management policies. Additionally, the creation of multidisciplinary decision-making oversight bodies on both political and regulatory levels could help prevent dysfunctional decision-making processes and extremist consensus positions.

#### 4.1.2. Self-Regulating Personal Resources and Skills

This theme pertains to fundamental, supportive personal resources and skills that should be fostered in As&R through career interventions. More precisely, in order to tackle the impact of contextual limitations on As&R and to counteract learned helplessness responses [18] research indicates that individual resources play a vital role by enabling an agentic perspective. For instance, factors such as general self-efficacy, resilience, entrepreneurial intentions [18], job search clarity, and hope-action competencies (hope, self-reflection, self-clarity, visioning, goal setting, implementing, and adapting) [24] have been found to support professional integration processes by fostering a hopeful career state and increasing work engagement [24].

Career adaptability is a key construct that has been studied in the reviewed research across a range of individual resources and contextual barriers. Referring to the definition of the construct outlined in the Career Construction Model of Adaptation [37], career psychology literature examining other population groups has found that this self-management skill, which includes four coping strategies (i.e., concern, control, curiosity, and confidence), leads to positive career transition outcomes. Research suggests that career adaptability in As&R operates as a mediator for both work self-efficacy and job search self-efficacy [17] and that it also mediates the relationship between psychological capital and confidence to engage in job search behaviours [19]. It has also been found that personal resources, such as resilience and self-efficacy, promote career adaptability and can be valuable psychosocial resources to facilitate As&R labour market integration [18]. Furthermore, in relation to future orientation and the definition of career goals, career adaptability is considered crucial, as it allows for setting and achieving long-term goals in relation to decent work [22]. Consequently, these studies consider this resource to be essential when As&R are confronted with challenging life and career transition contexts and suggest that it should be developed within career interventions targeting As&R. 

However, the key role of career adaptability in relation to other variables and its positive career outcomes in As&R is not consistently evident throughout the reviewed studies. In contrast to evidence from other target populations, the initial level of career adaptability of As&R does not seem to affect their job search attitudes and behaviours or occupational choices [17]. Similarly, enhancing general self-efficacy and job search clarity, as well as hope-action competencies (namely hope, self-reflection, self-clarity, visioning, goal-setting, implementation, adaptation, and job search clarity), led to higher work engagement, but no differences were found between the experimental and control groups in terms of career adaptability [24]. It is worth mentioning that the effect of career adaptability in As&R has been shown to vary according to contextual characteristics [19,25,29]. For instance, when confronted with social and administrative barriers in asylum countries, As&R with high psychological capital respond through increased career adaptability, leading to greater self-efficacy and determination in the job search. This effect is more intense in the interaction with administrative barriers, while it decreases in relation to social barriers [19]. The qualitative exploration of career adaptability in As&R in relation to the influence of contextual factors has shown that the latter varies across the four dimensions of career adaptability: The greatest contextual inhibiting effect has been noticed on the dimensions of control and confidence, whereas the effect on concern and curiosity was not on the dimensions themselves but on the behavioural level of exploring and realising career plans [23]. Also, with regard to the relevance of enhancing career adaptability in As&R to counteract their tendency to set and achieve materialistic present or proximal career goals [22], exploratory research showed that short-term planning was not associated with a lack of career adaptability in the dimension of concern, as As&R showed intense concern about their futures [23]. An additional aspect of career adaptability, as outlined in the McMahon et al. description of the construct [39], is cooperation. This aspect has been found to be significant in As&R, as they often belong to collectivist cultural backgrounds. Hence, the role of the interpersonal and relational nature of adaptive skills is highlighted [25].

Lastly, it should be noted that regarding the role of career adaptability and other individual resources examined above, the studies support their positive effects without testing whether perceptions of greater self-efficacy, used to measure positive career outcomes, are associated concretely with more and better job opportunities. Only one publication included a control group and both proximal and distal outcome measures in its design [24]. The study evidenced that improving hope-action competencies via a career intervention programme resulted in increased job satisfaction among As&R. Nonetheless, the experimental group did not experience greater job prospects or higher employability rates compared to the control group [24].

Further qualitative, quantitative, and longitudinal research is necessary to explore the role of career adaptability and other personal resources in the career transitions of the target population. It may be debated whether career adaptability is a precursor to true professional integration or whether, more likely, it involves greater acceptance of low-status, low-paid jobs to meet immediate survival needs: this may initially have a positive impact on subjective self-efficacy but may not necessarily correspond to decent work opportunities and real professional satisfaction and integration [50]. In light of the research reviewed, this core construct related to adaptive capacity should be seen not only as a personal trait requiring a focus on the individual within career interventions, but also as a dynamic process within systemic contextual interactions [25].

Additional resources identified by As&R through qualitative research include the following: personal values such as peace or freedom; personality traits, such as conscientiousness and extraversion [21]; physical, psychological, and moral courage strategies and resources [20]; and career interests [21]. With regard to the latter, practitioners should consider that As&R career interests do not develop in a vacuum but rather through their interaction with the social systems [51] and are limited by volition and access to opportunities [10,13]. Such interests are also associated with their cultural and social backgrounds, as well as gender roles, rather than being solely determined by autonomous decision-making processes generated by individual dispositions and orientations [21,26,27,31].

#### 4.1.3. Sense of Self and Identity in Career Transitions

This theme emphasises dynamic processes of identity negotiation and transformation, as well as the reconstruction of self-narratives, as core aspects to be addressed in career interventions in order to support the professional development of As&R in their career and existential trajectories. Work enables individuals to fulfil their basic needs for survival, social connection, and self-determination [52], while also cultivating a sense of usefulness and meaning that supports the creation of identity roots [53].

Most of the reviewed narrative research concentrates on how As&R construct, uphold, and alter their professional and social roles, sense of self, and identities during their career transition in the host country, and how they develop existential and professional orientations. The studies highlight dynamic processes that emerge from environmental interactions and result in ‘multi-layered’ self-perceptions that include both internal perspectives and external representations derived from direct social interactions and socio-political, historical, and cultural contexts [23,25,26,29,31,36].

The transition period into the host society presents As&R with an opportunity to negotiate a new identity, while the inability to continue in previous occupations leads to a loss of professional and social roles, impacting family life and self-esteem [23,26,28,29]. Social and professional barriers within the host community have been shown to undermine As&R identity needs for value, continuity, distinction, and control [29] and to inhibit adaptive behaviour [23]. To address these threats, As&R can adopt reactive-protective identity repair strategies, as well as proactive strategies such as self and emotion regulation, taking actions to change contexts, attending to their psychological growth, maintaining relationships with social and institutional networks, and engaging in the acquisition of new resources [21,29]. However, if unemployment persists over time, there is a considerable risk of developing identity crises, demotivation, and decreased self-esteem [54]. This shows particularly among As&R with highly qualified profiles, due to the centrality of their professional identification and the corresponding high levels of identity defence and conservative efforts, notwithstanding the inflexible constraints on the recognition of professional qualifications and competences [28].

Looking at gender differences, women’s identity transformation processes have been revealed to be more complex due to higher vulnerability related to experiences of violence and abuse; to cultural and social norms on gender roles prioritising family and childcare over career development [27]; and to stereotypical gender representations in the host country [26]. Career interventions need to make As&R women aware of these aspects and their career choices in relation to gender roles. In fact, As&R women’s professional identification and career choices are influenced by gender roles, which tend to orient them towards caring professions, while limiting their career choices to the most consuming and least remunerative jobs [21].

### 4.2. Design

This section allows us to clarify design aspects of the reviewed studies and includes reflections on the heterogeneity of the target population, research and career intervention settings, recommended approaches, methods, and tools in career counselling interventions.

#### 4.2.1. Heterogeneity of the Target Population

The participants in the analysed studies present different characteristics such as country of origin, cultural and religious backgrounds, and status; length of stay in the host countries; diversity of bureaucratic contexts and access to social and labour rights in the asylum country; historical and political circumstances that led to fleeing the country of origin; and heterogeneity of existential and career paths reflecting different personal resources and professional qualifications. These aspects demand special consideration in the design and implementation of career counselling.

As regards status, the reviewed studies do not address possible differentiations in the design of career interventions. This differentiation could be significant, as the legal status is directly linked to the ability to access social rights and services, which differ significantly depending on the protection status given and correspond to different needs.

#### 4.2.2. Setting

With regard to ‘setting’, understood here as the complex interplay of places, roles, and relationships, the studies were mainly conducted in settings in which As&R are usually located or linked to through asylum services. This choice offers the advantage of reaching out to a population that is difficult to access and facilitates the gathering of information and interaction with living contexts designed to reduce barriers to integration. The downside is that As&R may identify practitioners in psychology and researchers as part of the asylum system network and may react by restricting communication about experiences that vary from the personal narratives provided for the purposes of status recognition. Such responses stem from perceptions of strong power asymmetries vis-à-vis institutional actors, concerns about confidentiality and manipulative and unethical use of information [35,41,49] and fears of taking a critical or contrasting stance towards institutions due to repercussions on status and subsequent repatriation [35,41]. Such resistances are comprehensible when considering the institutional criteria for status recognition, which rely on the credibility and consistency of self-reported narratives [41,44,45] and the provision of services based on obedience and moral entitlement [41]. Finally, for many As&R, counselling may be alien to their culture, as intimate and difficult issues are normally handled within the family network, and mental health professionals are associated with psychological distress [35,49,53].

To help As&R overcome these resistances, it is important to clarify the independent position of counsellors and researchers in relation to the asylum system, as well as their commitment to confidentiality, ethics, and responsibility towards individuals and their communities [30]. Furthermore, counselling practitioners are required to reduce perceived inequalities [12,55,56] and establish a relationship of trust by, for instance, acknowledging counsellees as experts of their own existence [31,32,34,35]; by sharing personal experiences and values [30,32]; adopting an attitude of openness, curiosity, and respect for both As&R culture and their uniqueness [49]; and assuming a critical perspective to prevent promoting Western values and views [12,32,34,55,57,58,59].

#### 4.2.3. Approaches

In terms of career counselling approaches applied to the target population of this review, constructivist and socio-constructivist orientations were found to be effective as they focus on the subjective experiences of As&R, allowing them to integrate fragmented experiences of past and current career paths and proactively use their resources and skills [27,33,35,60].

The socio-cognitive [10,13,49] and systemic approach are also highly recommended frameworks for career counselling interventions with As&R, as they consider career development to be largely influenced by contextual factors. Specifically, the socio-cognitive Psychology of working theory model (‘PWT’) [10,52] recognizes a central role for context-related constraints on access to decent work opportunities. This model has recently been adapted to the target population [13] and provides an interesting theoretical framework for developing career counselling interventions.

In systemic career counselling models, As&R narratives of life stories are contextualised and reconstructed in relation to the complex interactions of life contexts and their influences on past, present, and future life. The systemic approach also allows the integration of different theoretical perspectives [11] that better respond to the specificities of the target population, i.e., the constructivist perspective [35] combined with the cultural perspective [57,58] and perspectives oriented towards social justice and critical consciousness development [12,55,56]. Therefore, the systemic perspective is particularly well-suited to career counselling with As&R. However, in the reviewed studies, the use of systemic theoretical frameworks in career counselling practice was very limited. Only three studies [25,31,36] referred to systemic theoretical models and tools or adopted a systemic perspective. To enable flexible models and tools for career counselling that address and respond to the complexities of intervention domains in ways that are inclusive of under-represented social groups such as As&R, career counselling practice needs more space for approaches based on socio-cognitive or systemic perspectives, in tandem with cultural and social justice orientations.

#### 4.2.4. Career Intervention and Tools

In addition to the most recommended and commonly practised individual career interventions, reviewed research also endorses group interventions, specifically the group-based model of existential career counselling [32]. This model offers the advantage of lower levels of personal exposure and uses similar interpersonal practices to those experienced in collectivist cultures, i.e., supportive and caring community interactions that counteract experiences of isolation, anxiety, and identity confusion. Moreover, the existential career counselling model for groups emphasises that individual needs and fulfilment within Western societies may not align with collectivist cultures [32]. Therefore, career counselling interventions should incorporate the role of family and community [25,31,61] for higher effectiveness.

Regarding career counselling tools and methods, studies particularly support the effectiveness of interventions that use narratives to enable As&R to explore their professional resources and constraints [21]; restructure self and life context representations and plan future actions [17]; reconstruct empowered self-narratives that counteract identification with suffering and trauma [20]; and situate their experiences within cultural, institutional, and social structures [31]. However, non-verbal forms of expression and graphic representation tools such as the ‘life journey’ [21] or the ‘life space map’ [33,35] as well as creative methods with plastic materials have also been recommended. Biographical exploration through visualisation is considered particularly effective and allows interventions to be more accessible, considering language difficulties and multiple resistances to counselling.

### 4.3. Limitations

Career interventions aimed at As&R represent an emerging field in career psychology, calling for further empirical research on a larger scale, including both longitudinal studies and comparative studies across asylum countries. Expanding future reviews to a greater quantity of studies will also enable more sophisticated analysis.

This paper has outlined relevant aspects to consider when designing and implementing career interventions with the target population. However, future research should consider expanding the number and range of databases and sources used. The use of databases primarily relevant to the field of psychology in this paper, and the exclusion of grey literature, both intended to ensure the relevance and quality of the studies reviewed, may have limited the number of eligible publications. Ultimately, this may have created a bias in the available findings.

Further limitations arise from the use of keywords referring to the target population. Future research might consider including keywords such as ‘migrant’ or ‘migratory background’. Indeed, terminology related to the wider concept of ‘migrant’ is frequently used in the analysed literature to encompass the category of the target population, even though ‘asylum seeker’ and ‘refugee’ populations differ from migrants in terms of their needs and require specific considerations with regard to the methods and aspects to be addressed in career interventions.

Similarly, concerning the disciplinary field, the research could be extended to include keywords such as ‘work’, ‘employment’, or ‘organisational well-being’, which are mainly studied in the field of Work and Organisational Psychology. A coordinated analysis encompassing both Career Psychology and Work and Organisational Psychology would provide a more comprehensive understanding of the refugee gap phenomenon and thus allow for better target career interventions.

Ultimately, it is worth noting that the studies reviewed include only a limited number of high-income asylum countries. Although such studies conducted in both European and non-European asylum countries have highlighted common research trends and similar features regarding migration and post-migration contextual constraints and resources, we strongly encourage country-specific career interventions. Furthermore, as the majority of As&R live in low- and middle-income countries, as reported by UNHCR [1], some recommendations in this paper may not be relevant to specific socio-economic and geographical contexts.

## 5. Conclusions

Despite the aforementioned limitations, this review allowed us to collect, discuss, and categorise the relevant aspects raised by the studies in relation to the research questions. Several relevant factors were identified with regard to the phenomenon of the ‘refugee gap’ under review. Determining factors included language barriers, limited social networks, inadequate recognition of professional skills and qualifications, and social and political contexts related to migration, including exposure to violence and abuse, restrictive political orientations, stereotypical public representations of As&R, degrading living conditions in the asylum countries, and feelings of powerlessness, suspension, and precariousness.

In light of the social and psychological challenges faced by As&R, career interventions constitute essential support for their professional and life transitions in asylum countries. As a result of the analysis of the existing empirical research on career development and counselling interventions targeting As&R, insights were gained into individual experiences and contextual factors that hinder their professional and social integration, while resources, strategies, and relevant processes of self-reconstruction were highlighted.

For the practical purposes of this paper, two schematic tools have been developed using a systemic perspective as an organising principle. The first maps binding factors and supportive resources related to the different system levels: the political-institutional and socio-cultural context, the social and community network system, and the system inherent to the individual. The second one provides useful guidelines for the design and implementation of career interventions, as well as recommendations for the competencies of career practitioners and researchers, and directions for ethical and caring practices towards As&R and their communities. It is important to consider that applying the insights gained from this review within career counselling interventions is subject to specific contextual influences, design considerations, and theoretical approaches. Yet, given the profound influence of political, institutional, and socio-economic factors in shaping the lives and careers of As&R, limited integration of such insights within personal agency-based models is likely to lead to an imbalance in counselling interventions towards individual resources and adaptive capacities, placing the responsibility on the individual for the difficulties that derive from complex dynamic systemic influences.

The first tool (see Figure 3)—a schematic representation of constraining factors and supportive resources at different systemic levels—can be used by practitioners both to define the design of career counselling and, during its implementation, to help explore and contextualise As&R experiences and to assist them in defining goals that are congruent with the dynamic and interrelated constraints imposed by different systemic contexts.

The second tool (see Table 4)—a set of guidelines for inclusive career counselling interventions—was developed from a combined review of the discussed key aspects identified in both career psychology literature and a broader psychosocial insight into the life systems of As&R. The recommendations are structured around the systems that most influence the career and life trajectories of As&R: the system of policy orientations and collective representation of As&R; the institutional system for the status determination; and the reception and integration system. For each domain, the initial two columns from the left outline the primary contextual factors and their effects. In response to each factor, in the third column, we provide concrete recommendations for a counselee-centred practice, the design features of counselling, the skills and role of the counsellor, and possible actions for career practitioners, assuming a transformative perspective that reconnects the discipline of career psychology to what has been its foundation since its inception: social commitment, solidarity, and ethical and social responsibility.

## Figures and Tables

**Figure 1 behavsci-13-00962-f001:**
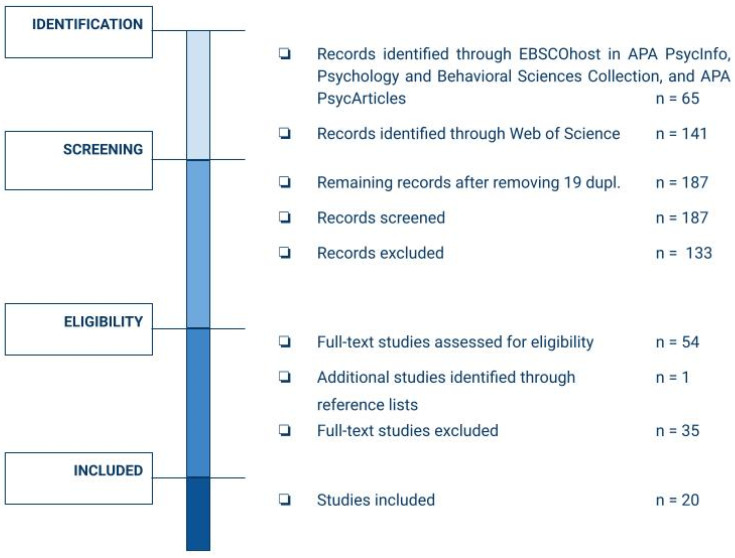
PRISMA flow diagram of systematic article search and selection.

**Figure 2 behavsci-13-00962-f002:**
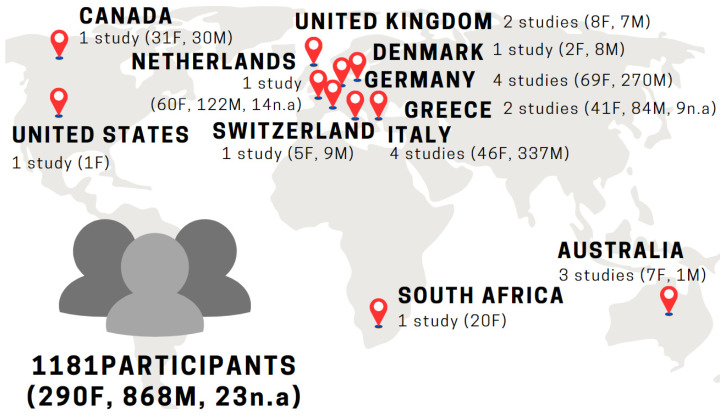
20 Studies in 11 countries (one study was conducted in two countries).

**Figure 3 behavsci-13-00962-f003:**
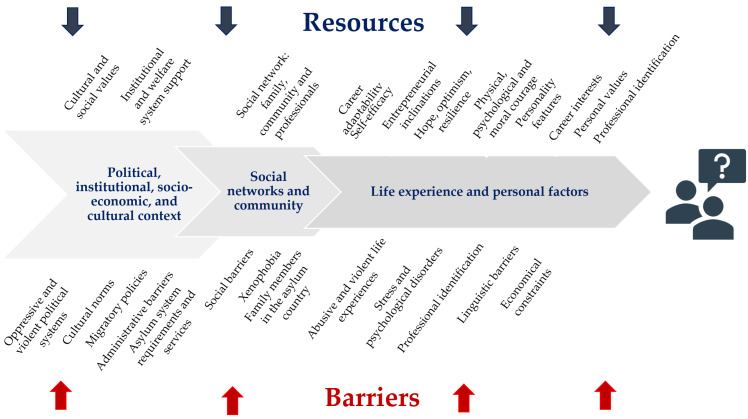
Resources and barriers to As&R professional development.

**Table 1 behavsci-13-00962-t001:** First group of studies.

Study Reference and Participants	Aim of the Study	Design/Methods	Main Findings
Morici et al., 2022 [17] Increasing refugees’ work and job search self-efficacy perceptions by developing career adaptability. Italy (Milan) 233 As&R (191 M; 42 F) Age ranged between 18 to 56 years African and Asian countries of origin	Explore the effects of a career counselling intervention in terms of greater job satisfaction, higher levels of career adaptability and perceptions of both work self-efficacy (WSe) and job search self-efficacy (JSSe).	Quantitative Experimental Administration of three scales: Career Adapt-Abilities Scale [37] (4 component scales: concern, control, confidence, and curiosity) 12 items on job search self-efficacy ‘The Perceived Job Search Self-Efficacy Scale’ (JSSe) (Farnese et al., 2007 in [17]) 10 items on work self-efficacy ‘The Perceived Work Self-Efficacy Scale’ (WSe) (Farnese et al., 2007 in [17]) Statistical analysis: pre- and post-intervention differences with Student’s *t*-test for paired samples; linear regression for testing the effects of career adaptability at predicting the growth of WSe and JSSe. No control group.	Participants were involved in a career counselling intervention in Italy (two individual interviews of one hour each and nine group interventions of three hours each). The three scales were administered one week before and after the intervention. After the intervention, all variables increased: The greatest increase was seen in the values of concern, curiosity and JSSe. There was also a medium-sized increase in the levels of control, confidence and WSe. WSe and JSSe values increased due to the growth of career adaptability, but their effect does not seem to depend on the starting level of career adaptability.
Obschonka et al., 2018 [18] Personal agency in newly arrived refugees: The role of personality, entrepreneurial cognitions and intentions, and career adaptability. Germany 267 As&R (M = 27.56 years; 78.1% male; 90% Syrians; 47.3% highly educated High employment rates in their country: 42.1% employed and 43% self-employed	Investigate the role of entrepreneurial intention and career adaptability as main personal agency indicators in facilitating the early transition process and the professional and social integration of newly arrived refugees in Germany.	Quantitative Experimental Measures: career adaptability [37] entrepreneurial intentions (Liñán and Chen’s, 2009 in [18]) Entrepreneurial alertness (subdimension: risk-taking (Wagner, Frick, & Schupp, 2007 in [18]); general self-efficacy (Hahn et al., 2016 in [18]) resilience (Montgomery, 2010 in [18]) Statistical data analysis: correlations and structural equation model.	Entrepreneurial inclinations, when fostered by self-efficacy and resilience, contribute to increasing skills of career adaptability. Self-efficacy and resilience predict entrepreneurial alertness, which mediates the link between these personality factors and career adaptability in newly arrived refugees. It is important for As&R in the early stages of integration to adopt an agentic perspective.
Pajic et al., 2018 [19] Antecedents of job search self-efficacy of Syrian refugees in Greece and the Netherlands. 330 Syrian refugees 40.6% settled in Greece and 59.4% in the Netherlands 30.6% F, 62.4% M, 7% undefined Average age about 32 years	Investigate the relationships among psychological resources, career adaptability, career barriers and job search self-efficacy.	Quantitative Experimental Administration of scales: psychological capital—PCPC-12 (Lorenz, Beer, Pütz, & Heinitz, 2016 in [19]): hope, resilience, optimism and self- efficacy career adaptability—CAAS (Savickas & Porfeli, 2012 in [19]) job search self-efficacy—JSSE (Saks, Zikic, and Koen’s, 2015 in [19]) career barriers: administrative and social barriers to labour market integration Statistical analysis: factor and path analyses; correlations; *t*-test to compare the two groups.	Participants in Greece had a higher average age and a lower level of professional qualifications and skills, compared to those in the Netherlands. In Greece, social barriers were perceived to be higher than in the Netherlands. Refugees in Greece had higher levels of career adaptability and self-efficacy in job search. Individuals with higher psychological capital are more confident in engaging in job search behaviours, due to their increased career adaptability. Yet, with higher social barriers this relationship weakened while with higher administrative barriers it strengthened.
Santilli et al., 2023 [20] Stories of courage in a group of asylum seekers for an inclusive and sustainable future. Italy 71 As&R (2 F, 69 M)	Giving asylum seekers a voice by exploring their stories of courage. Supporting them to design a sustainable future by promoting their positive assets.	Qualitative experimental. Snowball sampling of asylum seekers living in various reception centres. Administration of semi-structured interviews (inspired by Pury et al., 2007 in [20]). Thematic analysis of the interviews (recorded and transcribed) according to a circular inductive and deductive procedure.	Through their life stories, a greater understanding of contextual and personal barriers was gained. Classification of stories according to 3 forms of courage (moral, physical and psychological) and 3 life stages: pre-migration (freedom, decent life, decision to migrate, separation from family and country), migration (risk of death, degrading conditions, helping others, fear of the future) post-migration (working conditions or job interviews, fears about the future, trauma and guilt). Enhancing courage experiences as positive self- narratives and coping strategies.
Udayar et al., 2020 [21] Labour market integration of young refugees and asylum seekers: a look at perceived barriers and resources. Switzerland. 14 As&R from Eritrea and Somalia (5 F, 9 M) Age 19–25 years	Explore the perceived barriers and resources of newcomers (As&R) through a narrative career counselling intervention. Understand the impact of a state-level integration programme on the participants’ perception of their situation.	Qualitative experimental. Administration of a semi-structured interview (15 items) developed on the narrative metaphor of life’s journey (Denborough, 2014 in [21]) supported by a graphical tool that participants completed in itinere: a path representing life’s journey, a circle at the centre reflecting the present and a suitcase indicating resources. Data analysis: identification of the main As&R themes and visions cross-analysis of participant’s themes related to barriers and resources.	Main features emerging from the interviews. Barriers: linguistic and cultural; emotional (loneliness, nostalgia, unease due to living conditions); material (status uncertainty, access to labour market, financial difficulties and health problems of family members) Resources: social: family and friends in the home country, friends and family in Switzerland (resource and constraint), assisting professionals; personal: (5 types: qualities, career interests, work experience, activities, values); institutional: status, social rights.
Ginevra et al., 2021 [22] The role of career adaptability and future orientation on future goals in refugees. North Italy 75 (M) African As&R living in reception centres over 12 months	Explore refugees’ future goals and examine the relationship between career adaptability and future orientation.	Full mixed-method design and snowball sampling. Theoretical lens: Social constructivism—Life Design (Savickas et al., 2009 in [22]). Future goals: semi-structured interview ‘A Glance at the Future’. Qualitative data analysis to identify common categories of future goals. Career adaptability: Career Adapt-Abilities scale—short version (Maggiori et al., 2017 in [22]). Path analysis—relationship between the 4 components; Future orientation: Design My Future (Di Maggio et al., 2016 in [22]); Test: saturated model and covariances among variables and outcomes.	Qualitative data analysis showed that: participants tended to set materialistic goals rather than goals in terms of decent work. The main goals they mentioned were related to basic needs for survival and wellbeing, access to material resources and the need for affiliation. Secondary goals related to satisfaction of self-fulfilment needs were less frequently mentioned. Quantitative data analysis found that: Concern, control, confidence and curiosity directly and indirectly predicted the breadth of goals. Thus, career adaptability dimensions—mainly that of concern, control, and curiosity—help As&R to have a better future orientation. Recommendations: Practitioners should help As&R to develop career adaptability resources related to future orientation for the purpose of setting and achieving long-term goals related to decent work rather than focusing on short-term goals.
Wehrle et al., 2019 [23] Putting career construction into context: Career adaptability among refugees. Germany 36 As&R (6 F, 30 M), mostly from Syria (19) Mean age 32 years Employed in jobs or internships and resident for av. 3 years and 7 months	Exploring refugees’ career adaptation in the host country and the role of context in shaping adaptive responses.	Qualitative experimental. References to the Career construction theory [38] and to the four dimensions of career adaptability: concern, control, confidence, and curiosity [37]. Sampling: snowballing or via social service organizations. Data collection: Semi-structured interviews up to theoretical saturation. Data analysis: Coding and thematic analysis (NVIVO).	The ability of As&R to adapt their careers is strongly influenced by contextual barriers that hinder their agency. The impact of contextual factors on adaptive behaviour varied across the four dimensions of career adaptability: the greatest impact was on the dimensions of control (feeling helpless) and confidence (uncertainty and lack of self-confidence); the impact on concern and curiosity was not on the dimensions themselves but on being able to explore and realise plans. adaptive strategies: positive, appreciative attitudes (to take control and build self-confidence); setting clear career goals (concern) and persisting regardless of obstacles (control); making social connections; and exploring career dreams (curiosity).
Yoon et al., 2019 [24] The effect of a career development programme based on the Hope-Action Theory: Hope to work for refugees in British Columbia Canada 61 refugees (80% from Syria, 20% from Middle Eastern countries) Residents in British Columbia (Canada) for an average period of 12 months; searching for a job. At least 10 years of education, or 6 years and 4 years of work experience, able to commit to full-time employment.	Assess the proximal and distal outcomes of the ‘Hope to Work (H2W)’ career development programme for refugees based on the HAT (Hope Action Theory).	Quantitative Experimental 2 groups: experimental and control (t- student test). Test administration: before (T1), immediately after (T2), and 3 months after completion (T3) of the career intervention. Proximal outcome measures: hope action competencies (HAI; Niles, Yoon, & Amundson, 2010 in [24]); general self-efficacy (GSE; Schwarzer & Jerusalem, 1995 in [24]; career adaptability (CAAS-SF; Maggiori et al., 2017 in [24]); job search clarity (JSC, Zikic & Saks, 2009 in [24]). Distal outcome measures: employment status; career growth (CGS; Waters et al., 2014 in [24]); work engagement (UWES-SF; Schaufeli & Bakker, 2003 in [24]); hopeful career state; job satisfaction. Data analysis: covariance ANCOVA; correlation; serial mediation model.	The experimental group participated in the 2- week-long H2W programme, while the control group could only be referred to the employment career services. The programme included strengthening hope-action skills, visits to workplaces and 3 days of individual sessions to define an enhanced action plan for the development of HAT skills. The H2W programme was effective in increasing hope-action competencies, general self-efficacy, and job search clarity. There were no differences between the two groups with regard to career adaptability, the intensity of job-seeking activities, and the employment status or type of employment, but the work engagement of the experimental group was higher due to a hopeful career state.
Abkhezr, et al., 2022 [25] A systemic and qualitative exploration of career adaptability among young people with refugee backgrounds. Australia 5 young African people with refugee backgrounds (4 F, 1 M); Age: 20–28 Residents in Australia for 6–12 months in the framework of UNHCR resettlement programmes	Understand how career adaptability manifests in the participants’ lives and how it contributes to their career development after resettlement.	Secondary analysis of data collected through qualitative research. 2 semi-structured narrative interviews on life career stories and future career plans. Data analysis: thematic analysis based on 5 theoretical constructs of career adaptability [39]: concern, control, curiosity, confidence, and cooperation. deductive analysis using the systems of influences (STF) [11]; i.e., individual, social, environmental-societal, change over time, recursiveness, and chance).	The double level of qualitative analysis (thematic analysis and deductive analysis based on SFT) revealed that: the dimensions of career adaptability varied in relation to different contexts and at different times; the participants’ career development narratives reflected the iterative nature of the five dimensions of career adaptability as a result of the dynamic systemic influences. Recommendations: career adaptability in As&R should be considered as both a personality trait and a process that is dynamically reshaped within interacting systems and subsystems of influence.

**Table 2 behavsci-13-00962-t002:** Second group of studies.

Study Reference and Participants	Aim of the Study	Design/Methods	Summary/Main Findings
Berr et al., 2019 [26] Broadening the understanding of employment and identity of Syrian women living in Germany: A biographical study. Germany 5 Syrian women with refugee status and high professional qualifications Age: 25–55 years Residence of at least 2 years in Germany, enrolled in UNHCR resettlement programmes	Understanding the interplay between forced migration, employment, and identity. Giving refugee women a voice and contributing to overcoming stereotyped images about them. Exploring the professional identity construction processes of Syrian women in relation to their life stories and life conditions after migration.	Qualitative experimental Biographical narrative interviews. Thematic and conversation analysis according to the ‘reconstruction of narrative identity’ (Lucius-Hoene and Deppermann, 2004 in [30]). Analysis of three different levels of narration: temporal (life stages) social (individual-environment interaction in different life contexts). self-referential (self-awareness of identity profiles and positioning).	Professional histories were collected, and identity processes were explored. Five life themes were identified and the interaction between them: Education as a socio-economic resource (access to society and achievement of high social status in Syria) Significance of related persons (role models, collective career orientations and decision making) Career development expectations vs. reality Professional (re)orientation in Germany (fight against stereotypes and access to education and employment) Future employment prospects (stress).
Nyabvudzi & Chinyamurindi, 2019 [27] The career development processes of women refugees in South Africa: An exploratory study South Africa 20 As&R women from African countries	Explore the career development processes of women refugees in the host country and the main influences and structural constraints to their career and lived experiences.	Qualitative experimental adopting a feminist, intersectional and career constructivist theoretical lens. Snowball sampling. In-depth narrative interviews on the career development of the participants. Data were audio-recorded, transcribed and e-mailed to the participants for accuracy. Thematic and content analysis.	Main findings from the data analysis: Women shifted their focus from the long-term perspective to the short-term perspective of physiological needs; Socio-economic constraints, gender and its overlap with family roles and the pressure due to xenophobic attitudes influence career development and lived experiences. Gender, xenophobia and structural barriers intersect and overlap.
Mackenzie Davey & Jones, 2020 [28] Refugees’ narratives of career barriers and professional identity. United Kingdom. 15 persons with refugee status; professionally qualified in medicine and teaching	Explore the identity processes of refugees with high qualifications and status in their home country while facing barriers to the recognition of qualifications and professional status in the UK. Investigate how identity tensions between past, present, and future professional experiences are managed.	Qualitative experimental Interviews and focus groups on career barriers, plans and future aspirations. Inductive thematic analysis.	Participants were subjected to narrative interviews and took part in focus groups discussing the obstacles they encountered in their career path, career planning and future aspirations. Attempting to restore a coherent self-narrative, participants implemented actions to recover previous professional identities and to build new ones. Professional identification and identity work were found to be both a supporting and hindering factor for refugees with former high career status.
Wehrle et al., 2018 [29] Can I come as I am? Refugees’ vocational identity threats, coping, and growth. Germany (resident for more than 2 years) 31 As&R (7 F, 24 M), mostly from Syria (23), average age 28 years, employed in jobs or internships	Understand the impact of primary contextual threat sources on vocational identities of As&R and their integration into society.	Qualitative experimental. Snowball sampling. Semi-structured interviews up to theoretical saturation. Coding and thematic analysis (NVIVO). Codes: partly informed by the literature on identity threats, coping mechanisms and psychological growth and partly derived from the data.	Participants stressed the importance of work as a financial and self-defining means. The main contextual threat sources were insecurity, language, social exclusion and external stereotypical attributions. These factors threaten the basic human needs for a sense of worth, distinctiveness, continuity, and control. Coping responses consisted of protection or restructuring identity strategies, but also proactive attitudes to prevent threats.

**Table 3 behavsci-13-00962-t003:** Third group of studies.

Study Reference and Participants	Aim of the Study	Design/Methods	Main Findings
Abkhezr et al., 2020 [30] Exploring the boundary between narrative research and narrative intervention: implications of participating in narrative inquiry for young people with refugee backgrounds. Australia 5 young African people with refugee backgrounds (4 F, 1 M) Age: 20–28 Residents in Australia for 6–12 months—UNHCR resettlement programmes	Exploring the boundaries between a narrative career counselling intervention and qualitative narrative inquiry research. Exploring career development through the narratives of life career stories.	Qualitative experimental. Social constructionist perspective. Semi-structured narrative interviews of 70–90 min. Test FCA (The Future Career Autobiography; Rehfuss, 2009, 2015 in [30]): i.e., writing a text about one’s professional future, before the first interview (T1) and after the second interview (T2). Inductive thematic analysis of interviews. Comparative narrative analysis on the FCA texts in T1 and T2.	The comparative analysis of the FCA revealed changes in participants’ future career plans. While career counselling interventions require explicit change, participation in narrative research does not. These impacts should be considered when designing narrative research. Focus points on the researcher’s role: Quits the expert role and adopts active and responsive listening strategies with an open and curious attitude Pays attention to the power imbalance between researcher and participant, engages in self-reflection processes and shares reflections or personal experiences Acts in a responsive and responsible way adopting ethical and caring practices towards participants and their community (no detached attitude).
Abkhezr et al., 2018 [31] Finding voice through narrative storytelling: An exploration of the career development of young African females with refugee backgrounds. Australia 3 African women with refugee backgrounds living in Australia UNHCR resettlement programmes	Explore the career development processes of 3 young women with refugee backgrounds through a careful exploration of the cultural and contextual features of their career stories.	Qualitative research: exploratory multiple case study. Semi-structured narrative interviews on life career stories. Data analysis: social constructionist approach based on a voice-centred relational analysis—VCRA (Brown & Gilligan, 1993; Gilligan et al., 2003; Mauthner & Doucet, 1998 in [31]): listening for the plot; listening for ‘I-poems’; listening for relationships; placing participants within their cultural contexts and social structures.	Narrative career counselling proved useful by enabling participants to re-contextualise their skills, strengths, knowledge, and future career plans. This gives space to voices that may have been lost and enhance a greater sense of agency for implementing future career plans. Each participant’s career story reflects the interaction between various voices, relationships, social structures, and dominant narratives that influence career plans. Voices create different relationships with other inner voices, those of others or those representing specific social or cultural contexts.
Petersen et al., 2022 [32] Existential career guidance for groups of young refugees and migrants: A Danish initiative. Denmark 10 young migrants and refugees from the Middle East Average age 17 years	Apply and investigate the effects of a five-step intervention model of existential career guidance with a group of young refugees and migrants.	Action research project—career intervention Implementation of the existential career guidance model for groups: 10 group sessions for a 6-month period. Thematic analysis of the empirical data: 9 semi-structured group interviews (3 with counsellors and 6 with participants, before, during, and after the intervention).	The existential career counselling group intervention consisted of ten 2.5-h sessions in which participants reflected on their life experiences aimed at building a meaningful life. The intervention proved effective in developing a greater sense of belonging, feeling understood and respected, having self- awareness, hope and the willingness to care for others, as well as improving language skills. By employing self-disclosure, counsellors gained a deeper acceptance of the participants, modified their prejudices and activated self-reflective skills.
Pierce & Gibbons, 2012 [33] An ever-changing meaning: a career constructivist application to working with African refugees. United States 26-year-old female Somali refugee, mother of 2 children, ages 4 and 6	Illustrate how to use narrative constructivist career counselling to address career-related issues with African refugees.	Application of a constructivist narrative career counselling through a case study. Intervention phases: co-construction—the counsellor supports life and career narratives using the ‘life space map’; de-construction—identification of themes, personal and cultural values. Definition of professional values; construction—development of career goals, construction of future stories and planning (means and strategies).	Recommendations: the counsellor’s attitude of empathy, warmth, acceptance and genuineness supports the awareness of the meaning-making processes and their influence on life choices; focus on career rather than trauma; cultural awareness, sensitivity and mindfulness of potentially encouraging Western values; mitigate the power differential that As&R may perceive due to their migration experiences; integrate the constructivist approach through advocacy and case management.
Schultheiss et al., 2011 [34] Career, migration and the life CV: A relational cultural analysis. 3 As&R United Kingdom	Understand the career path of migrants and As&R and explore alternative approaches to effective career guidance and counselling interventions for As&R.	Participatory arts and action research/workshop: application of a career intervention model through the creation of a ‘life CV’. Career intervention approach: integration between the relational cultural paradigm (Schultheiss, 2007 in [34]) focused on the intertwined nature of relational and working life and the Life Design model (Savickas et al., 2009 in [34]). Workshops involving refugees, researchers, and artists in 3 small groups to complete a personalised ‘life CV’ through narratives and artistic means, such as poetry, photography and drawing.	The life CV process allows one to examine As&R employability potential within the context of their biographies and life experiences, including their past, their journeys, and their current situations. This approach appears to be effective in improving the understanding of competences, abilities and skills and in giving voice to experiences and visions of the future It facilitates the understanding of the impact of historical, political and cultural contexts on identity. The construction of meanings related to the whole life supports the development of new perspectives, life projects and identities.
Słowik, 2014 [35] ‘Life space mapping’ as an innovative method in career counselling for refugees, asylum seekers and migrants. As&R Greece	Implementation of a multicultural narrative career counselling model.	Applied theoretical study—presented during a workshop for professionals and students of psychology during a conference in Greece. A narrative career counselling model based on constructivist life design, a multicultural approach and the graphic tool of the ‘life space map’ (Peavy, R.V., 1997 in [35]). The map consists of drawing three circles (past, present and future) and a ’ladder’ (a rainbow) to represent the future steps and tasks (actions, changes).	This career counselling model supports As&R in constructing meanings and future planning and facilitates contact and awareness of stereotypes and barriers. The approach places significant emphasis on cultural aspects and utilises visualisation tools to reduce anxiety and resistance that AS&R may experience due to cultural and living experiences as well as language difficulties. Visualisation and plastic materials support the process of biographical exploration and stimulate self-analysis of both needs and solutions. Counsellors ought to cultivate non-verbal communication proficiency and intercultural awareness, embrace a curious and open-minded approach, and utilise active listening methods.
Magnano et al., 2022 [36] Approaches and strategies for understanding the career development needs of migrants and refugees: The potential of a systems-based narrative approach. Italy 4 participants: 2 women from Romania and 2 men—from Pakistan and Cameroon	Highlight how narrative career counselling supports individuals in activating meaning-making processes.	Qualitative research/Intervention study–analysis of case studies. Interviews conducted by counselling experts based on the MSCI (My System of Career Influences) maps of the participants (McMahon, Watson, & Patton, 2013 in [36]). 2 levels of analysis: Themes and relationships—focused on systems of influences. Meaning attributed to systems of influences through the Voice-centred relation analysis (Gilligan et al., 2003 in [36])	Participants shared their experiences using different perspectives, emphasizing the impact of their own internal voices or external voices from others, as well as social, cultural, and historical influences. The findings highlight that career development takes place in different social systems. Narrative systemic counselling supports the development of a comprehensive and cohesive understanding about the influence of life contexts and systems on personal experiences, as well as on social and work integration for As&R.

**Table 4 behavsci-13-00962-t004:** Guidelines for inclusive career counselling interventions.

**System of Policy Orientations and Collective Representations of As&R**					
**Contextual** **Factors**	**Impact**	**Assisting Counsellee’s Processes**	**Career Counselling Approaches and Methods**	**Recommendations for Career Counsellors**	**Advocacy and Interventions at the Socio-Political Level**
Restrictive migration policies based on control and repression.	Dangerous migratory routes (dehumanisation, abuse and violence). ‘Natural selection effect’ of borders: wealthier, more resilient and skilled As&R reaching the Global North.	Critical analysis of the influence of the socio-cultural and political systems. Supporting identity processes (defensive and proactive strategies).	Targeted intervention according to the specific legal and social status in each asylum country. Reconstruction of narratives within socio-cognitive or systemic frameworks in combination with cultural, critical consciousness and social justice perspectives.	Curious and open-minded attitude. Multidisciplinary expertise for contextualisation (geopolitical, social, and cultural contexts). Recognizing prejudices and cultural differences. Critical consciousness on Western cultural hegemony and dominant ideologies. Reduce counsellee resistance and mistrust through empathic listening and a commitment to confidentiality.	Enhance the political agency of psychology. Form bodies to monitor political and regulatory decision-making processes. Draft reports and psychosocial analyses on the consequences of migration management legislation and practices. Define common guidelines applying to category, care, and representation associations. Tackle social bias caused by political discourse.
Disparities between countries with regard to the standard of services and access to social and labour rights.	Difficult post-migratory paths (degrading living conditions). As&R experiencing low self-esteem, powerlessness, threat to basic identity needs and humiliation.				
Representation of As&R as suffering victims and survivors.	Impact on As&R: induced identification with suffering stories reactivation of painful experiences, lack of social agency, ‘learned helplessness’ Experiences of mistrust and rejection, or denial of parts of oneself.	Valuing positive narratives of courage and resilience. Enhancement of individual resources (resilience, adaptability, courage and coping strategies). Voicing internal parts and depowered narratives.	VCRA: exploration of interacting voices, relationships, social structures and dominant narratives. Biographical exploration through nonverbal forms of expression (graphic and artistic means).	Address traumas objectively, without drawing excessive attention to them. Counter power inequalities: show empathy, warmth, acceptance; act on prejudices; share personal experiences.	Avoid using the disclosure of As&R suffering as a means of advocacy. Reassert the social and political origins of migratory and post-migratory suffering in all psychosocial studies and reports.
**Institutional System for The Status Determination**					
**Contextual** **Factors**	**Impact**	**Assisting Counsellee’s Processes**	**Career Counselling Approaches and Methods**	**Recommendations for Career Counsellors**	**Advocacy and Interventions at the Socio-Political level**
Indefinite waiting times for status recognition procedures.	Risk of developing psychological disorders with lingering effects.	Strengthen psychological capital, self-efficacy, career adaptability. Support protective, proactive, and restructuring identity processes. Planning and defining professional and social networking goals.	Approaches: Constructivist, socio-constructivist, socio-cognitive (working on agency and decision-making processes). Asylum system and service network coaching. Creating ‘life CVs’.	Multidisciplinary skills to identify systems of institutional power and bureaucratic procedures. Empathetic and curious attitude. Trust in the client and renunciation of the expert role. Reduce resistance and power discrepancy: independent settings from asylum services; empathetic and responsive to trauma (if arises); use of graphic tools; ev. existential group counselling. Conscious and responsible use of narrative tools	Research and reports on the impact of the suspension and indefinite duration of the procedure on the physical and psychological health of As&R.
Autobiographical interviews (doubts about the credibility and consistency of memories in status recognition procedures).	Attitude of decision-makers: discretion in decisions, distrust, and rejection of authenticity of life stories, avoidance of trauma, low level of empathy, cynical attitudes and power asymmetry.	Stimulating visions of the self as a multiplicity and not as a single, circumscribed, and objective unity. Turning attention to the interpretative processes of the self.	Systemic approaches and VCRA analysis. Using alternative or complementary tools to autobiographical narration (e.g., life space map, graphic and expressive tools).		Research on the psychophysical impact of the evaluation and questioning procedures of asylum state bodies. Create guidelines for conducting hearings in an ethical and gender-sensitive manner.
**Reception and** **Integration System**					
**Contextual** **Factors**	**Impact**	**Assisting Counsellee’s Processes**	**Career Counselling Approaches and Methods**	**Recommendations for Career Counsellors**	**Advocacy and Interventions at the Socio-Political Level**
Management of the reception system: humanitarian logic superimposed on state supervision.	It reinforces control, repression, and merit on the basis of suffering and weakness. It compels As&R to adopt attitudes of gratitude and conformity.	Stimulating freedom of expression and identifying critical issues in the system. Voicing internal parts and valuing positive narratives of courage and resilience	Attention to the setting within reception centres or participation referred by institutional services. Explain the duty of confidentiality and ethics of career counselling.	Build relationships of trust. Stimulate freedom of expression with respect to critical factors and reassure about the lack of repercussions. Take professional and ethical responsibility towards As&R and their community. Place emphasis on the drive for autonomy and awareness of the impact on the lives of As&R (coordinated actions and interventions with services). Consider the peculiarities of the host country’s labour and social rights provisions.	Extend research and counselling to people outside the reception circuits and services. Research on the dynamics with institutional services and the influence on psychological well- being of As&R.
Social and professional integration services.	Impact on As&R: dependency on services, limitation of autonomy, risk of marginalisation, power asymmetry, perceived lack of control and influence on systems and services, bias in interventions (perceived lack of freedom of expression and omission of critical issues).	Strengthening skills, psychological capital and increasing career adaptability and self- efficacy in job search and work. Develop a critical understanding of services and labour market and share reflections and critical experiences	Individual interventions with group modules. Exploration of meaningful and positive narratives of the self. Support the construction of long-term perspectives. Create ‘life CV’.		Include As&R in social and professional integration programmes. Carry out research in areas outside the reception services and study the underground economy networks in which persons who have been denied a status end up: trafficking, child trafficking, prostitution and drugs.
Linguistic, social, and economic barriers and obstacles regarding qualifications and skills recognition.	Powerlessness, perceived lack of recognition, continuity and control.	Develop a contextualised reading ability of experiences and counteract identification with suffering and powerlessness.			Participate in working groups to promote policy actions for skills and qualifications recognition.

## Data Availability

Not applicable.

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
