# Peer review of "Taking Action towards an Inclusive Career Counselling for Asylum Seekers and Refugees—A Literature Review Based on the PRISMA Model"

_behavsci, 2023, doi:10.3390/bs13120962_

Round 1

Reviewer 1 Report

Comments and Suggestions for Authors

The proposed paper is a careful and thorough review of studies on career counseling for asylum seekers. The paper adds value to the existing literature both from a theoretical and practical point of view by proposing concrete recommendations at the end of the paper. I congratulate the authors for their work and effort. I make a few minor comments below, mainly related to the techno-editing:

1. There are some typos in Table 1

2. Correct to mdpi reference system line 209 

3. The number of Figure 3 must be corrected to Figure 2

4. The recommendations Table must be properly numbered and named 

Author Response

Thank you very much for your encouraging comment.

We edited the paper and addressed your requests

  1. There are some typos in Table 1 – Thank you, done.
  2. Correct to mdpi reference system line 209 – Thank you, done.
  3. The number of Figure 3 must be corrected to Figure 2. - Thank you, done.
  4. The recommendations Table must be properly numbered and named  - Thank you, done.

Reviewer 2 Report

Comments and Suggestions for Authors

The Introduction section could be edited somewhat to clarify language and set up the rest of the paper. Since a large portion of the paper is a table laying out the merits of each of the included articles, some set-up is required to offer the reader how best to read and understand the material they are about to read. 

Comments on the Quality of English Language

Most of the revisions needed are between lines 31-41. This paragraph reads awkward and minor edits would allow for the flow of reading to be easier. 

Author Response

Thank you very much for your comments that helped us to improve the quality of this work.

- The Introduction section could be edited somewhat to clarify language and set up the rest of the paper. Since a large portion of the paper is a table laying out the merits of each of the included articles, some set-up is required to offer the reader how best to read and understand the material they are about to read. 

Thank you for this feedback. We added a paragraph in the introduction (lines 76-92) explaining the structure of the article for the purpose of helping the reader. Furthermore, we have expanded the paper to include more references that help contextualizing the findings with insights from career psychology and multidisciplinary research (previously 47 references, now 62 references).

- Comments on the Quality of English Language

Most of the revisions needed are between lines 31-41. This paragraph reads awkward and minor edits would allow for the flow of reading to be easier. 

Many thanks for your feedback. The paragraph has been reedited (lines 36-41).

Reviewer 3 Report

Comments and Suggestions for Authors

The method has been sufficiently described, but has some limitations that will be discussed below. The literature search was limited to two international and big databases, the found publications have partly very different methodological approaches, target groups (age, countries of origin, countries of destination etc.) and relevant context. Perhaps the inclusion of further databases to extend the literature pool should be considered. The summaries and review of the literature reviewed are comprehensive and contain the relevant information.

With regard to the guiding question and objectives of the article "to identify effective career counseling and vocational guidance interventions to address the widespread phenomenon of the refugee gap. […] Besides the practical value of preventing conditions of severe social distress for As&R, the identification of such interventions would also contribute to the theoretical development of career psychology"; and the formulation of conclusions and recommendations for action, however, a number of problems arise:

There is a lack of a systematic approach in the processing of the results, which includes the strongly varying study designs and contexts: There is a lack of explanation of how these very different studies are dealt with in the following analysis and recommendations for action.   

The systemic perspective which considers individuals as open systems is very interesting, but this approach was not sufficiently included in the discussion, the elaboration of the findings and formulation of recommendations of action.

The conclusions and recommendations are based on a partly very limited number of studies (partly single studies).

In order to strengthen the results and thus the validity of the conclusions and recommendations for action, a restriction to certain study designs or at least a clear differentiation of the different results should be made. In addition, it would be worth considering limiting the geographical area to e.g. Europe.

A description of the further handling of the results of the systematic literature search, i.e. the targeted and differentiated analysis and interpretation with regard to due question and objectives, are not sufficiently elaborated at the beginning of the article.

A comprehensive description of the limitations is missing and needed.

Comments on the Quality of English Language

Many typos that still need to be corrected, mainly in the tables.

Author Response

Thank you very much for your comments that helped us to improve the quality of this work.

-The method has been sufficiently described, but has some limitations that will be discussed below. The literature search was limited to two international and big databases, the found publications have partly very different methodological approaches, target groups (age, countries of origin, countries of destination etc.) and relevant context. Perhaps the inclusion of further databases to extend the literature pool should be considered. The summaries and review of the literature reviewed are comprehensive and contain the relevant information.

Thank you for the feedback. Following this suggestion, 2 new databases were included. We screened a total of 187 studies and 20 were ultimately eligible. Major changes in the text are indicated in red. In the paper, we have also specified some clarifying information regarding the selection of databases which are particularly relevant to the field of career psychology.

We have also stated that this review is not intended to be exhaustive. It has outlined some of the most relevant aspects covered by the empirical literature in an expanding field of intervention in career psychology, and that future research could be further extended by using more databases and keywords, as well as complementary disciplinary areas in psychology (see section 4.3 on limitations of the review).

- With regard to the guiding question and objectives of the article "to identify effective career counseling and vocational guidance interventions to address the widespread phenomenon of the refugee gap. […] Besides the practical value of preventing conditions of severe social distress for As&R, the identification of such interventions would also contribute to the theoretical development of career psychology"; and the formulation of conclusions and recommendations for action, however, a number of problems arise:

Many thanks for the suggestion. The formulation of the paragraph has been revised in line with the research questions (lines 109 – 114) and the considerations addressed in the discussion and in the final recommendations (both revised).

- There is a lack of a systematic approach in the processing of the results, which includes the strongly varying study designs and contexts: There is a lack of explanation of how these very different studies are dealt with in the following analysis and recommendations for action.   

Many thanks, these aspects have been made explicit in the methodology section (lines 152-166) and in the limitations section (section 4.3). Additionally, the conclusion section has been updated with information on the creation of these tools, their possible use for career practitioners, and their intent to address the practical purposes of the research questions.

-The systemic perspective which considers individuals as open systems is very interesting, but this approach was not sufficiently included in the discussion, the elaboration of the findings and formulation of recommendations of action.

We have tried to better clarify and make this explicit in the introduction section (lines 76-92), in the discussion section (lines 256-266) and in the conclusions (lines 618-655).

In the article we refer to this perspective and specifically to the systemic levels as a framework and criterion for both analysing and categorising knowledge emerging from the studies. Additionally we highlight how individual experiences are influenced and interact with macro- and meso-level life context. The same framework and criterion has also been used in developing and presenting the practical tools and recommendations. These are grouped according to the systems of influence, offering for each one an overview of the effects on individual experiences and considerations on how such effects can be addressed from three complementary angles: intervention with the client and methodological means in career counselling; the practitioner's skills and attitudes; and ethical safeguarding actions that psychologists and the field of career Psychology as a whole are expected to provide outside of the intervention setting.

-The conclusions and recommendations are based on a partly very limited number of studies (partly single studies).

Even after extending the research, the number of studies remains limited and in section 4.3 of the discussion (limitations) we point this out as a bias.

-In order to strengthen the results and thus the validity of the conclusions and recommendations for action, a restriction to certain study designs or at least a clear differentiation of the different results should be made. In addition, it would be worth considering limiting the geographical area to e.g. Europe.

In the section on the limitations of the review, we have added some considerations on the limitations related to geographical areas (see lines 592-599). However, regarding the suggestion to limit the recommendations to European countries only, we believe that this does not reflect what emerges from the analysis. Although non-European studies are under-represented, common themes relating to the contexts of influence between European and non-European countries have been highlighted. Furthermore, the three identified research strands, which reflect the aspects to be targeted by interventions, comprise both European and non-European studies. We have made this explicit with some disclaimers in the discussion, limitations and conclusions section.

We acknowledge that the limited number of studies, due to the emerging research field and other restrictions, did not allow for a more complex analysis (see section 4.3). Nevertheless, distinguishing between study designs was considered inappropriate for the purposes of the research, as several studies address common constructs and aspects with both qualitative, quantitative and mixed methods research (e.g. career adaptability).

- A description of the further handling of the results of the systematic literature search, i.e. the targeted and differentiated analysis and interpretation with regard to due question and objectives, are not sufficiently elaborated at the beginning of the article.

Thank you. This was addressed in the introduction (lines 81-92)

-A comprehensive description of the limitations is missing and needed.

Thank you. Section 4.3 of the discussion includes a description of the main limitations of the study.

Comments on the Quality of English Language

Many typos that still need to be corrected, mainly in the tables.

Thank you, tables have been revised.  

Round 2

Reviewer 3 Report

Comments and Suggestions for Authors

Thank you for this interesting article. The revisions are extensive and detailed. I wish you continued success in this field of research.